# Breast Reconstruction following Mastectomy for Breast Cancer or Prophylactic Mastectomy: Therapeutic Options and Results

**DOI:** 10.3390/life14010138

**Published:** 2024-01-18

**Authors:** Laurentiu Simion, Ina Petrescu, Elena Chitoran, Vlad Rotaru, Ciprian Cirimbei, Sinziana-Octavia Ionescu, Daniela-Cristina Stefan, Dan Luca, Dana Lucia Stanculeanu, Adelina Silvana Gheorghe, Horia Doran, Ioana Mihaela Dogaru

**Affiliations:** 1Department of General Surgery, “Carol Davila” University of Medicine and Pharmacy, 050474 Bucharest, Romania; laurentiu.simion@umfcd.ro (L.S.); rotaru.vlad@gmail.com (V.R.); sinziana.ionescu@umfcd.ro (S.-O.I.); luca_dan94@yahoo.com (D.L.); horia.doran@umfcd.ro (H.D.); 2General Surgery and Surgical Oncology Department I, Bucharest Institute of Oncology “Prof. Dr. Alexandru Trestioreanu”, 022328 Bucharest, Romania; 3Zetta Hospital, 020311 Bucharest, Romania; inapetrescu@yahoo.com; 4Department of Oncology, “Carol Davila” University of Medicine and Pharmacy, 050474 Bucharest, Romania; cristina.stefan10@gmail.com (D.-C.S.); dana.stanculeanu@umfcd.ro (D.L.S.); adelina.silvana.gheorghe@gmail.com (A.S.G.); 5Oncology Department I, Bucharest Institute of Oncology “Prof. Dr. Alexandru Trestioreanu”, 022328 Bucharest, Romania; 6Surgical Clinic I, Clinical Hospital Dr. I. Cantacuzino, 030167 Bucharest, Romania; 7Department of Plastic Surgery, “Carol Davila” University of Medicine and Pharmacy, 050474 Bucharest, Romania; ioana.m.dogaru@gmail.com; 8Department of Plastic Surgery, Emergency University Hospital, 050098 Bucharest, Romania

**Keywords:** breast reconstruction, reconstruction following mastectomy, prophylactic mastectomy, chest feminization, transgender, implant reconstruction of breast, immediate reconstruction, delayed reconstruction, two-stage breast reconstruction, autologous breast reconstruction

## Abstract

(1) Importance of problem: Breast cancer accounted for 685,000 deaths globally in 2020, and half of all cases occur in women with no specific risk factor besides gender and age group. During the last four decades, we have seen a 40% reduction in age-standardized breast cancer mortality and have also witnessed a reduction in the medium age at diagnosis, which in turn means that the number of mastectomies performed for younger women increased, raising the need for adequate breast reconstructive surgery. Advances in oncological treatment have made it possible to limit the extent of what represents radical surgery for breast cancer, yet in the past decade, we have seen a marked trend toward mastectomies in breast-conserving surgery-eligible patients. Prophylactic mastectomies have also registered an upward trend. This trend together with new uses for breast reconstruction like chest feminization in transgender patients has increased the need for breast reconstruction surgery. (2) Purpose: The purpose of this study is to analyze the types of reconstructive procedures, their indications, their limitations, their functional results, and their safety profiles when used during the integrated treatment plan of the oncologic patient. (3) Methods: We conducted an extensive literature review of the main reconstructive techniques, especially the autologous procedures; summarized the findings; and presented a few cases from our own experience for exemplification of the usage of breast reconstruction in oncologic patients. (4) Conclusions: Breast reconstruction has become a necessary step in the treatment of most breast cancers, and many reconstructive techniques are now routinely practiced. Microsurgical techniques are considered the “gold standard”, but they are not accessible to all services, from a technical or financial point of view, so pediculated flaps remain the safe and reliable option, along with alloplastic procedures, to improve the quality of life of these patients.

## 1. Introduction

Breast cancer accounted for 685,000 deaths globally in 2020, and half of all cases occur in women with no specific risk factor besides gender and age group. During the last four decades, we have seen a 40% reduction in age-standardized breast cancer mortality [1] and have also witnessed a reduction in the medium age at diagnosis, which in turn means that the number of mastectomies performed for younger women increased, raising the need for adequate breast reconstructive surgery. Advances in oncological treatment have made it possible to limit the extent of what represents radical surgery for breast cancer, yet in the past decade, we have seen a marked trend toward mastectomies in breast-conserving surgery-eligible patients [2]. Prophylactic mastectomies have also registered an upward trend [3,4]. This trend together with new uses for breast reconstruction like chest feminization in transgender patients [5] has increased the need for breast reconstruction surgery.

Breast cancer is the most commonly diagnosed neoplasm in the female population in the world [6]. It is the leading cause of cancer-related death in women in most countries of the world, except in developed countries, where it ranks second after lung tumors. However, mortality has been steadily declining for over 30 years, with an average 5-year survival of 86% and 75% at 10 years [7]. This trend is attributed both to the increase in the effectiveness of oncological treatments and to early screenings and screening programs similar to those for other neoplastic diseases [8,9].

Breast reconstruction is an important component of breast cancer treatment. With the increase in life expectancy, it has become essential to ensure a good quality of life for patients, forcing a continuous evolution of surgical techniques. Breast reconstruction is necessary not only after performing a modified radical mastectomy, but also after conservative interventions on the breast that have not been accompanied by an optimal aesthetic effect. The need to complete the surgical treatment of breast cancer with breast reconstruction derives from the beneficial impact at the psychological level, respectively, for the body image, sexuality, and general quality of life of patients [10]. In recent years, the ever-increasing number of patients opting for prophylactic mastectomies due to a genetic predisposition for developing breast cancer or a family history of cancer [3,4,11,12,13] has given birth to a new type of integrated treatment plan in oncology. Changes in guidelines, prompting the genetic testing of BRCA mutations and the availability of those tests even in the absence of an oncologist’s recommendation, have determined an increase in the number of women getting tested and then opting for a contralateral or bilateral prophylactic mastectomy. A good example for this trend is the Angelina Jolie effect on the Western population; following the known actress’s double prophylactic mastectomy, there was a noticeable increase in the number of healthy women requesting this procedure and having it performed.

Although it is not the focus of this study, breast reconstruction has had another extremely important purpose in recent years, namely, for the chest feminization of male-to-female transgender patients. In combination with hormone and psychological therapy, breast enhancement is the most common physical modification in this populational subset [5,14,15,16], contributing to a reduction in the patient’s dysphoria. For this purpose, all surgical reconstructive techniques used in patients with mastectomies can be employed.

This study reviews the main techniques, especially the autologous and mixed procedures, and investigates available data from the literature, indicating their indications and results.

## 2. Problem at Hand

### 2.1. Dimension of Problem

Breast cancer accounted for 685,000 deaths globally in 2020, and half of all cases occur in women with no specific risk factor. During last four decades, we have seen a 40% reduction in age-standardized breast cancer mortality [1], but we have also witnessed the reduction in the medium age at diagnosis, which in turn means that the number of mastectomies performed for younger women increased, raising the need for adequate reconstructive surgery. Breast cancer is the most prevalent form of cancer in the world, with a total of 7.8 million women alive in 2020 who were diagnosed with a form of this malady in the previous 5-year period [1]. Advances in oncological treatment, which have prolonged patients’ survivorship after breast cancer, have also made it possible to limit the extent of what represents radical surgery for breast cancer, yet in the past decade, we have seen a marked trend toward mastectomy in breast-conserving surgery (BCS)-eligible patients [2]. Prophylactic mastectomy has also registered an upward trend [3,4]. This trend, together with new indications for breast reconstruction like chest feminization in transgender patients [5,14,16] or the need to resolve the asymmetry of the contralateral breast [17,18,19], have increased the need for breast reconstruction surgery. 

### 2.2. Mastectomy vs. Breast-Conserving Surgery 

Many breast cancer patients elect to have a radical mastectomy, rather than a conservative surgical procedure, even though they are perfect candidates for BCS, and some long-term studies have suggested a slightly more favorable outcome of lumpectomies associated with radiotherapy vs. mastectomies [20], maintaining the high frequency of mastectomies. Patients opting for mastectomies over BCS usually do not choose by taking into account the histology, localization, or aggressiveness of the tumor, but rather more subjective reasons like a lack of trust that BCS can offer the same likelihood of cure as a more extensive procedure [21] or fear of additional procedures. The surgeon’s recommendation is a key factor in the decision-making process, but it is overshadowed by the patient’s fear of cancer [22]. 

In recent years, an increase in mastectomy rates in early-stage breast cancer patients was observed. The reasons for which patients tend to select a more aggressive procedure when breast-conserving surgery is an option are unclear and include, besides a so-called “peace-of-mind” and a more laxed surveillance schedule, the easy access to reconstructive surgery and the patient’s confidence in the aesthetic results of reconstructive techniques [23].

### 2.3. Lymphadenectomy and Sentinel-Lymph Node Biopsy

Radical surgery for breast cancer comprises the excision of the tumor (mastectomy or various breast-conserving techniques) and a procedure addressed to the axillary lymph nodes (inferior lymphadenectomy, extensive lymphadenectomy, or identification and excision of the sentinel lymph nodes using radioactive material or intravital dyes like Indocyanine green or Methylene Blue). The extension of the excision of the lymphatic tissue can influence the results of the reconstruction of the breast in both immediate and delayed settings by increasing the number of complications. Complete axillary lymph node dissection has a more pronounced effect when compared to that of a limited lymphadenectomy of a sentinel lymph node excision and is associated with a greater probability of implant loss independent of the associated radiotherapy [24,25]. There are studies that proved that the excision of each node increases the risk of reconstructive surgery complications by 4% [25]. The same study concluded that the removal of four or more lymph nodes can adversely affect the immediate reconstructive procedure by seroma formation or even implant loss [25]. However, the complications after immediate reconstruction of the breast are associated with the use of implants. For this reason, in patients requiring axillary lymph node dissection, the oncoplastic surgeon should offer the autologous methods of reconstruction [24]. 

Half of the patients with mastectomies for breast cancer elect to undergo reconstructive surgery [26,27] due to aesthetic considerations and an improved quality of life [28,29,30,31] through reduced body dysmorphia. Yet, following reconstruction, many patients experience sequelae like functional limitations of the upper limb (strength and mobility) and pain [31,32,33]. Axillary lymphadenectomy can cause neurological syndromes like pain, paresthesia, and limitations of mobility after reconstructive surgery. This effect can be reduced by preserving the sensitive nerves during the lymphadenectomy [34].

### 2.4. Impact of Radiotherapy in Surgical Options and Results

After reconstructive surgery, radiation therapy may affect the aspect of the operated breast, including the altered skin color and rigidity. It can also lead to capsular contraction, which mandates the removal of the implant. Patients undergoing radiation therapy after reconstructive surgery need to be advised about the possibility of additional corrective surgery [35].

### 2.5. Quality of Life following Surgery for Breast Cancer

After mastectomies, patients report a significant alteration of their quality of life (QoL) through a series of mechanisms: body dysmorphia affecting both emotional and sexual functioning, especially in younger patients [36]; pain and limited mobility in the ipsilateral upper limb; and psychological effects like negative emotions such as sadness, low mood, and dejection [37]. Although there is a significant reduction in the alteration of QoL following immediate breast reconstruction, many women tend to underestimate the impact of the mastectomy and to be overly optimistic about the impact of reconstructive surgery, and a significant proportion of them (up to 20% in some studies) come to regret breast reconstruction [36,38]. This particular aspect needs to be taken into account when discussing breast reconstruction with the patients in order to make sure they have realistic expectations.

When discussing QoL in patients who underwent mastectomies, we cannot leave out the problems caused by breast asymmetry, especially in large breasts, leading to alterations of the skeletal system (like scoliosis). Asymmetry of the breasts can occur even in patients with reconstructive surgery after their mastectomy if the procedure was unilateral and performed with implants. The remaining breast tends to be more ptotic, resulting in undesired aesthetic effects and causing the patient to request corrective surgery of the contralateral breast. 

### 2.6. Oncologic Follow-Up and Results after Reconstructive Surgery

Breast reconstruction surgery is a safe procedure from the oncological point of view, regardless of using an autologous or implant-based method for reconstruction and regardless of an immediate or delayed timing of the procedure [39], and it does not increase the local or systemic recurrence rates nor disease-free and overall survival [40,41]. The type of reconstructive surgery after a mastectomy does not influence the recurrence rate independently of the aggressive histology of the tumor [42], lymphatic invasion, and the positive resection margins [43].

Another safety concern after breast reconstruction following a modified radical mastectomy is the possibility of the detection of recurrence, since autologous tissue below the skin flap may interfere with the detection of recurrent nodules, and fatty necrosis can confuse the diagnosis [44,45]. Recurrence after reconstructive surgery using implants may be challenging to detect beneath the implant [46]. A thicker skin flap (over 1.5 cm) may interfere with the detection of a palpable mass upon examination, and an extremely slim skin flap (under 0.5 cm), although conducive to an early clinical detection of recurrence, is more prone to necrosis of the flap. A delicate balance needs to be achieved, and in our opinion, a 1 cm thickness of the skin flap is optimal. Lastly, in order to minimize the risk of flap necrosis, techniques using Indocyanine green may be employed for assessing the perfusion of the flap used for breast reconstruction [47].

## 3. Breast Reconstruction

### 3.1. Timing of Breast Reconstruction—Immediate or Delayed

Breast reconstruction is classified by the type and time of surgery. Immediate reconstruction takes place at the same time as the mastectomy, and secondary (or delayed) reconstruction is performed from a few months to several years after the mastectomy. Currently, it is performed at least three months after the end of radiotherapy and generally at about a year after the mastectomy [48]. The two main types of reconstruction are with implants or autologous tissue; they can also be used together in mixed procedures.

Immediate breast reconstruction (Figure 1) has certain benefits over the secondary one, especially in terms of patient satisfaction, quality of life, and psychological status post-mastectomy [49]. These patients are relatively more protected from the psychological effects of mastectomies, and studies have shown a stable evolution of quality of life and satisfaction of this group compared to patients receiving delayed reconstruction [50,51]. In the latter case, the quality of life is significantly improved with the reconstructive procedures, with the results ultimately being equalized in the long run [52]. Also, after the immediate reconstruction, a more natural and aesthetic result is obtained, with the intervention usually being associated with a skin-sparing mastectomy which respects the inframammary groove and keeps the skin intact, proven safe from the oncological point of view by a series of studies, provided the correct selection of patients [53,54,55,56]. An important factor for selection is the appreciation of the thickness of the skin flap, this being correlated with the aesthetic results and with the possible postoperative complications [57]. Patients undergo fewer major surgeries and require fewer days of hospitalization, recovering faster postoperatively. From an oncological point of view, immediate reconstruction is considered safe and has been shown to not increase the risk of local recurrence compared to that of mastectomies without reconstruction [58,59]. At the same time, this technique does not change the effectiveness of adjuvant radiotherapy [60].

Despite the many benefits of immediate reconstruction, many surgeons choose to postpone the operation for another time (Figure 2), often for reasons of oncological safety. Definite diagnoses of malignancies of radiologically detected breast tumors are made more and more frequently by guided biopsies, and the real extension of the tumor tissue can be evaluated macroscopically only intraoperatively, and microscopically only when examining the mastectomy piece. Thus, the subsequent therapeutic attitude is often decided intraoperatively [61]. However, in order to evaluate the quality and thickness of the skin flap intended for immediate breast reconstruction, mammography, breast ultrasound, and magnetic resonance imaging (MRI) can be used preoperatively to complete the clinical examination. The results of these investigations guide the decision on surgery for immediate breast reconstruction and have been shown to be true to intraoperative findings. The thickness of the flap is important in choosing the type of implant used, but also for avoiding postoperative complications such as skin necrosis [57].

The indications of radiotherapy, typically applied to patients at high risk of recurrence (>4 positive lymph nodes or positive resection margins), tend to increase, with studies proving its usefulness in patients with 1–3 positive lymph nodes [62,63,64]. Although radiotherapy does not contraindicate immediate reconstruction, the higher rate of complications, especially in implant-only reconstructions, is a second reason why in these patients, either a two-stage reconstruction or a delayed reconstruction is chosen [65]. After reconstructive surgery, radiation therapy may affect the aspect of the operated breast, including altered skin color and rigidity. It can also lead to capsular contraction which mandates the removal of implant. Patients undergoing radiation therapy after reconstructive surgery need to be advised about the possibility of additional corrective surgery [35]. 

Another contraindication of immediate reconstruction is any modifications of the flaps (tegument and subcutaneous tissue), namely, the presence of necrosis, inflammation, or signs of dermal neoplastic dissemination resulting in a large skin defect after the mastectomy.

### 3.2. Two-Stage Breast Reconstruction

In 2002, the technique of two-stage breast reconstruction was initially described for delayed reconstruction. Later, the technique was especially used to improve the results in cases associated with radiotherapy [66]. The ionizing radiation used on either the chest wall or the axilla irreversibly alters the tissues in the irradiated field, regardless of their nature. In the short term, erythema and scaling of the skin can appear, and in the long-term, severe fibrosis, telangiectasia, hyperpigmentation, and tissue atrophy [67]. Under these conditions, many surgeons prefer to place a tissue expander at the time of the mastectomy, preserved during radiotherapy, which aims to maintain both the shape and the skin needed for the final reconstruction [68]. The expander can be filled at the time of the intervention, or progressively, depending on the condition of the flaps and the center where the intervention is carried out [69]. It can be partially emptied before radiotherapy sessions in order to favor the alignment of the irradiation fields, but this step is not always necessary [70]. Subsequently—it is recommended no later than 3 months after the completion of radiotherapy—the second stage of reconstruction is performed, usually with autologous tissue. For patients who do not require adjuvant radiotherapy, the recommendation is that the second stage of reconstruction be performed no later than two weeks after the mastectomy [65].

### 3.3. Breast Reconstruction with Implant

Regarding the type of intervention, at present, it is estimated that 80% of breast reconstructions are performed with an implant [68]. This type of intervention is shorter and easier from a technical point of view, and postoperative recovery is faster.

In the long run, however, complications are more common than in cases of breast augmentation (30% at 5 years compared to 12% at 5 years) and are accentuated by the history of radiation therapy [69,70]. The main complications are capsular contracture, implant rupture, hematoma, and infections [71]. Implant reconstruction is associated with aesthetic complications like asymmetry, chest wall deformity, mispositioning or displacement, ptosis, wrinkling or rippling (wrinkling of the implant that can be felt or seen through the skin), skin rush, redness and bruising, and inflammation. The implant can suffer deflation, rupture, or extrusion. Many of these complications will require additional surgeries, a possibility that the patient should be informed about. Seroma, hematoma, delayed wound healing, infection, and necrosis of the skin/flap can also occur after implant breast reconstruction. These complications will require additional treatment and will most often delay adjuvant therapies with effects on the overall oncologic outcome. Following infection, hematoma formation, and seroma formation, capsular contraction can occur. Grade III and IV capsular contraction (hardening of the breast around the implant, causing painful tightening of the tissues) will require corrective surgery, but could occur again after the procedure. Implants are associated also with more exotic complications including other cancers; there have been reports of Breast Implant-Associated Anaplastic Large Cell Lymphoma (BIA-ALCL), squamous cell carcinoma, and mesenchymal tumors after breast reconstruction with implants.

However, implant reconstructions remain preferable for many surgeons because they avoid the complications of the donor areas and generate lower costs, and in the absence of radiotherapy or in a two-stage reconstruction, they are a simple solution with good aesthetic results. Acellular dermal matrices (ADMs) are used either as a first intervention to support the implant in the lower pole, not covered by the pectoralis major, or in reinterventions [72]. These biological materials are made from human, bovine, or porcine dermis processed to remove all cellular components—which can generate an immune response—and keeping the extracellular matrix containing mainly collagen (85%) along with proteoglycans, glycosaminoglycans, and elastin, arranged in a network, in the meshes of which the host cells are arranged [73]. This integration of the matrix provides good support for the breast prosthesis and a high-quality capsule, resulting in a natural appearance of the final reconstructed breast. The high costs are the main disadvantage and make dermal substitutes inaccessible on a large scale. With the increase in accessibility, it is estimated that the approach to implant reconstructions in a single stage will change significantly.

### 3.4. Autologous Breast Reconstruction Techniques

Autologous techniques are considered by many authors to be the gold standard in breast reconstruction. They consist in restoring the contour and volume of the mammary gland with the help of either rotating, pediculated flaps, which retain their vascular source, or micro-surgically freely transferred flaps from other areas of the body, most often from the abdomen. The intervention can be performed immediately or delayed, like the implant reconstruction. Moreover, if the volume provided by the flap is not sufficient, other techniques such as the free transfer of autologous fat (lipofilling) or placement of an implant may be associated [74].

The advantages of flap reconstruction.

Flap reconstruction offers several advantages, including improving the quality of irradiated tissue by bringing healthy tissue into a scar area; the final appearance after reconstruction is a natural one that mimics, in time, the physiological ptosis of the contralateral breast, does not require reinterventions for replacement after a period of time, can be used in patients who do not want or do not tolerate an implant, and is the recommended type of reconstruction for the radio-treated patients [75].

b.Types of transferred free flaps.

First described in 1989 by Koshima and Soeda [76], the freely transferred flap based on the inferior epigastric artery (DIEP) has long been the preferred alternative in autologous breast reconstruction in specialized centers [76]. Other free flaps that are described but rarely used in practice are the TRAM (transverse rectus abdominis myocutaneous flap), more commonly used in its pediculated version; TUG (transverse upper gracilis flap); SGAP (upper gluteal artery perforator flap); or IGAP (lower gluteal artery flap). Lower-limb flaps are indicated in selected cases, in the absence of a suitable abdominal donor area or in patients with previous interventions at this level [77].

c.Latissimus dorsi pediculated flap.

First described by Tansini for covering chest wall defects in 1906, the latissimus dorsi pediculated flap began to be used in breast reconstruction after almost 70 years [78]. Until the middle of the twentieth century, the radical mastectomy technique described by Halsted recommended either grafting or secondhand healing of the resulting defect, strongly contraindicating any form of reconstruction, as it was considered to “hide possible recurrences and promote the spread of tumor cells” [79].

The evolution of oncological treatments, a better understanding of the pathology, and the increase of patients’ life expectancy, together with the appearance of breast implants, changed the approach of these cases. Schneider and Botswick described in 1977 and 1978, respectively, the latissimus dorsi flap accompanied by the implant in restoring the physiological contour and ptosis of the breast after a mastectomy [80,81]. Subsequently, Papp and McCraw modified the flap, including subcutaneous adipose tissue overlying the muscle in order to achieve implant-free reconstruction [82].

Although it is no longer the gold standard in autologous reconstruction, the reliability and predictability of its anatomy still make it preferred by many surgeons for delayed reconstructions and also the preferred rescue option in the event of free-transfer flaps failure [83]. Currently, its primary uses are in patients who do not have sufficient reserves for a free flap; those with a personal history of abdominal interventions; or those with significant comorbidities such as obesity and diabetes or in smoking patients [84].

The most common complications are seromas in the donor area, usually easy to treat without further intervention. Associated with alloplastic procedures, capsular contracture has been described more frequently in association with implants and less frequently in two-stage reconstructions, when the implant is preceded by an expander. Rare complications are contour defects in the donor area, limited shoulder mobility, and decreased muscle strength in the arm and the scapula alatae [85].

In Figure 3, Figure 4 and Figure 5, we present various immediate or delayed reconstructive techniques using the autologous or mixed procedures we employed for our patients.

## 4. Discussion

Breast surgery has rapidly evolved in parallel with oncological treatments; while in 1970, the safety of reconstruction after a mastectomy was still questioned, today it is suggested for most patients who want it, so today, the task of choosing the most appropriate technique for each case is on the shoulders of the surgeon. With all the options available, the surgeon chooses the right technique by taking into account his own experience and preference; available resources and factors related to the patient, such as the breast size to be reconstructed, skin quality, type of mastectomy indicated, disease stage, adjuvant treatments, surgical history, and the general condition of the patient; and last but not least, her preference.

The contraindications of reconstruction are relatively few, limited to patients with a precarious general condition which do not allow an elective intervention as well as cases with a definite unfavorable life prognosis, which do not justify additional interventions. Also, patients with unrealistic expectations about the end result or who do not accept postoperative scars are not good candidates for reconstruction [86]. Age is no longer considered a contraindication to either the procedure itself or the choice of surgical technique, although, for reasons beyond the general condition and possible associated diseases, techniques involving freely transferred flaps are not usually recommended in patients over 65 years [48].

The selection of oncological surgery, tumorectomy, or mastectomy, as the case may be, contributes significantly to the end result. The decision on whether or not to preserve the mammary gland in early cases is still a matter of debate. A study by Veronesi et al. [87] following the development of 700 women with tumors < 2 cm for 20 years showed that breast preservation interventions (tumorectomies/lumpectomies) do not change the long-term survival when compared to mastectomies, although the local recurrence rate is higher in the first situation; Morrow and co-workers [88] also showed that for stages 0-II, a third of patients end up requiring a mastectomy. The American Society of Breast Surgery has recommended breast preservation whenever possible, along with the association with adjuvant oncological treatments such as chemo- and radiotherapy [89]. However, more recent data from the United States show an increase in the preference for mastectomies, especially prophylactic, in patients with and without BRCA 1/2 mutations [90].

The long-term benefit of this radical gesture has been demonstrated in cases with the presence of mutations, or in familial cases, in studies such as that performed by Boughey et al. [91], which followed a group of 385 women with a family history and stage I or II tumors and found that after 17 years, survival was significantly improved in patients with bilateral mastectomies. Meanwhile, another study by the same author [92] showed that bilateral mastectomy increases hospitalization costs and the number of on-call visits in the first 2 years, recommending that these data be explained to patients before making a decision.

Hoskin et al. [93] conducted a study in the USA on 3195 women operated on for breast tumors over a period of 5 years, between 2009 and 2014. Of the patients who required mastectomies, the proportion of patients who opted for immediate reconstruction increased by 31%. The percentage of prophylactic bilateral mastectomies with immediate reconstruction increased by 20%, while for the same intervention without reconstruction, the percentage decreased by 10%, from 22 to 12%.

Complications after intervention are not significantly different between tumoral and healthy breasts, but in the case of bilateral procedures, the complication rate increases significantly compared to that of unilateral ones, from 6.3% to 10.6%, according to some authors [94], respectively, and from 4.2% to 7.6%, according to other studies [95], this aspect being one of the main criticisms of this trend.

Statistics on the incidence of breast cancer in Romania are limited. The existence of a national patient record that would include, among other things, the stage at the time of diagnosis would contribute to the understanding of epidemiology and would facilitate a unified, multidisciplinary approach and faster access of patients to treatments. From the experience of oncological surgery centers, many patients with breast cancer who present for treatment are detected to be in advanced stages locally, with larger tumors and often with clinical or radiological lymph node involvement. This situation significantly changes the surgical indications and, implicitly, the reconstructive options. Although surgical excision is sometimes possible primarily through the radical mastectomy technique, patients usually receive neoadjuvant chemotherapy. Given the stage of the disease, reconstruction in such cases is most often delayed until the completion of oncological treatments [96]. However, the evolution in the diagnosis and treatment of breast cancer has led to the development of oncoplastic surgery that not only allows for the preservation of the breast, but also obtains better aesthetic results in oncological safety conditions [97].

A number of studies have evaluated the safety of immediate breast reconstruction in neoadjuvant-treated patients with favorable results. A meta-analysis conducted in 2020 by Varghese et al. [98] evaluated 17 observational studies, comprising 3429 cases, and revealed that it does not increase the risks of perioperative complications such as hematoma, seroma, or difficult wound healing and does not delay adjuvant treatment. The study instead showed a lower rate of complications in younger patients, as well as a higher rate of complications in patients who smoke or have a high body mass index. Also, patients with large breasts (>600 g) had a higher complication rate. Neoadjuvant chemotherapy slightly increases the risk of complications related to implants or expanders and insignificant risks related to autologous procedures, as noted by the same authors.

The effect of adjuvant chemotherapy on the results of reconstruction is difficult to estimate, as most patients also benefit from radiation therapy during treatment. One study showed a relative risk of liponecrosis of 4.8 in cases where immediate reconstruction with a free flap was performed [99].

Radiotherapy can significantly affect the postoperative outcomes, especially in alloplastic procedures. Reconstruction using autologous procedures is safer, with a lower rate of complications. El-Sawabi [100] performed a meta-analysis on complications after breast reconstruction in irradiated patients and showed that autologous procedures are associated with a lower rate of post-procedural complications (wound healing, seromas, hematomas, infections, and reinterventions) when compared to implant-based reconstruction (30.9% vs. 41.3%) [100]. Failure of the intervention occurred in 16.8% of alloplastic procedures and only 1.6% of autologous ones. When radiotherapy was performed on the temporary device, the complication rate was higher than when it was performed on the permanent implant (18.8% and 14.4%, respectively).

Among the autologous procedures, the latissimus dorsi myocutaneous flap has long been the basic choice for reconstruction, associated or not with an implant. Almost any patient can benefit from this technique due to the reliability and versatility of this flap. The main controversies are related to the transferred volume, the aesthetic result, and the secondary functional deficit of the shoulder and arm.

As early as 1986, Russel and colleagues [101] observed that although there is a decrease in scapular girdle muscle function immediately postoperatively and this effect may be more evident in athletic or elderly patients, this deficit does not have a significant impact on daily activities—except in athletes, skiers, swimmers, and climbers—and fades in about 6 months due to the development of synergistic musculature. The muscle strength has been showed to be comparable to preoperative levels in 3 months (2015 study by Yang) [102]. However, this can also be associated with the neurologic alterations following the sectioning of the sensitive nerves during axillary lymphadenectomy; this can be mitigated by the utilization of a modified surgical technique for lymphadenectomy which preserves the intercostobrachial nerve and the third and fourth intercostals [34].

On the subject of aesthetic results of the reconstruction after a mastectomy, Lindegren [98] conducted a study on secondary autologous-type reconstructions with 70 irradiated patients comparing the perceptions of both patients and surgeons of the aesthetic results after using the latissimus dorsi flap or DIEP. Although the surgeons favored the DIEP due to the natural shape and volume of the reconstructed breast, the patients were more satisfied with the latissimus dorsi flap reconstruction. This result was unexpected for the authors, which they hypothesized was correlated to higher satisfaction in the latissimus flap with the scar of the donor area [103]. Another study had the opposite results in a larger group of patients but a small percentage of irradiated patients [104].

The appropriate volume for larger breasts can be recreated either by a combination with the implant, by serial lipofilling sessions, or by changing the skin palette to include more subcutaneous adipose tissue [105].

Breast reconstruction has become a necessary step in the treatment of most breast cancers, and many reconstructive techniques are now routinely practiced. Microsurgical techniques are considered the “gold standard”, but they are not accessible to all services, from a technical or financial point of view, so pediculated flaps remain the safe and reliable option, along with mixed and alloplastic procedures, to improve the quality of life of these patients.

## Figures and Tables

**Figure 1 life-14-00138-f001:**
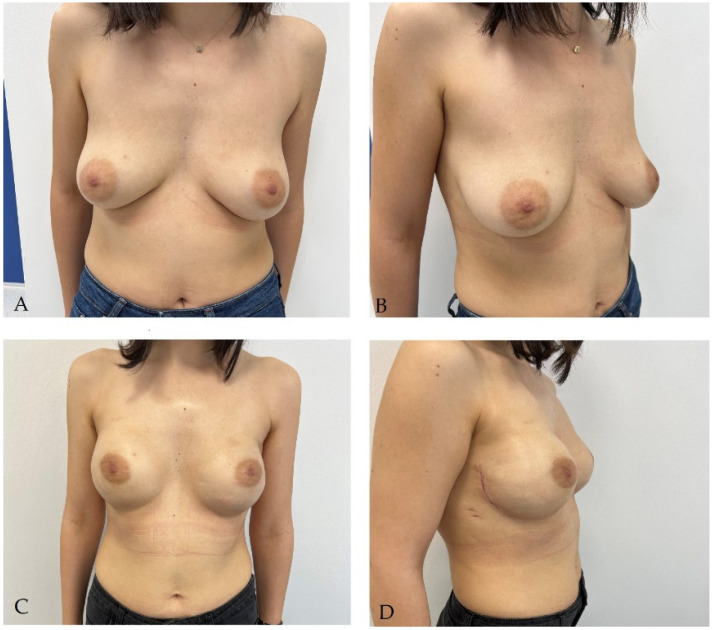
Patient with stage Ia invasive ductal carcinoma of left breast and BRCA positive status. She underwent bilateral subcutaneous mastectomy with left sentinel lymph node identification using Indocyanine green followed by immediate bilateral reconstruction with 350 cc round implants: (**A**,**B**) aspect before reconstructive surgery; (**C**,**D**) aspect at 3 months after reconstructive surgery.

**Figure 2 life-14-00138-f002:**
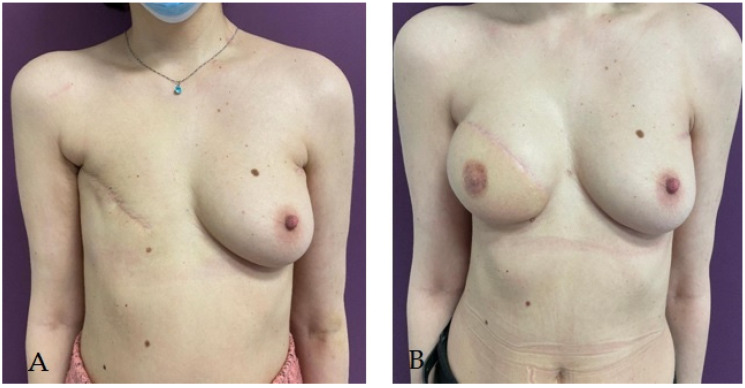
Patient with right radical mastectomy for breast cancer followed by radiotherapy; she underwent right breast delayed reconstruction using latissimus dorsi pediculated flap and a 225 cc round implant. (**A**) aspect before reconstructive surgery; (**B**) aspect at 3 months after reconstructive surgery.

**Figure 3 life-14-00138-f003:**
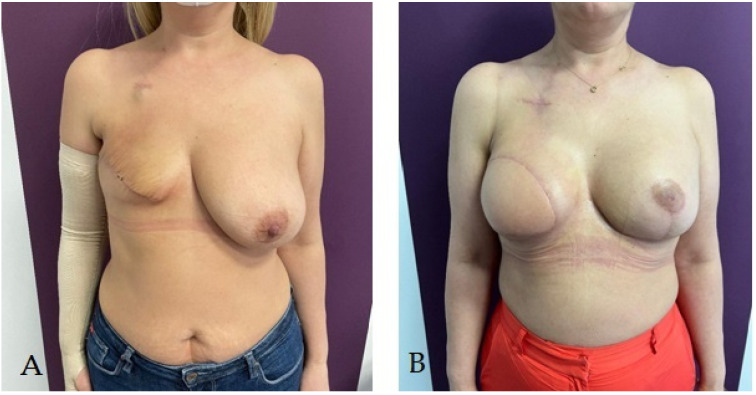
Patient with right radical mastectomy for breast cancer followed by radiotherapy; she underwent right breast delayed reconstruction using latissimus dorsi pediculated flap and a 320 cc round implant simultaneous with prophylactic left subcutaneous mastectomy (due to BRCA-positive status) with immediate reconstruction using a pediculated inferior dermoadipous flap and a 350 cc round implant: (**A**) aspect before reconstructive surgery; (**B**) aspect at 3 months after reconstructive surgery.

**Figure 4 life-14-00138-f004:**

Bilateral prophylactic mastectomy in patient with BRCA-positive status and heavy family history of breast cancer. Immediate reconstruction using 325 cc round implant and pediculated inferior dermoadipous flap followed by nipple–areola complex graft: (**A**) aspect before reconstructive surgery; (**B**) aspect at 3 months after reconstructive surgery; (**C**) final aspect at 32 months after reconstructive surgery.

**Figure 5 life-14-00138-f005:**
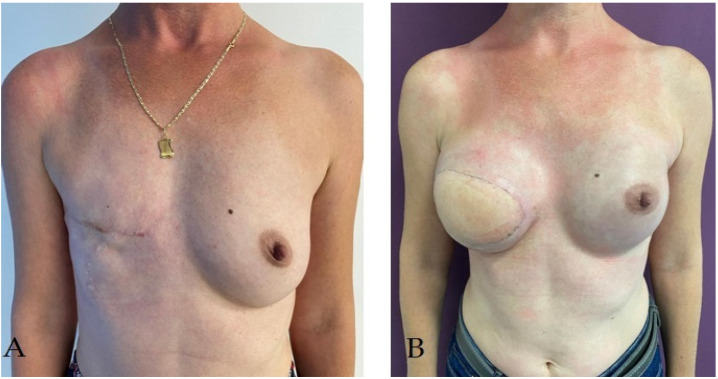
Patient with right radical mastectomy for breast cancer; she underwent right breast delayed reconstruction using latissimus dorsi pediculated flap and a 275 cc round implant simultaneous with prophylactic left subcutaneous mastectomy (due to BRCA-positive status) with immediate reconstruction using a 325 cc round implant: (**A**) aspect before reconstructive surgery; (**B**) aspect at 3 months after reconstructive surgery.

## Data Availability

Not applicable.

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
