# Peer review of "Breast Reconstruction following Mastectomy for Breast Cancer or Prophylactic Mastectomy: Therapeutic Options and Results"

_life, 2024, doi:10.3390/life14010138_

Round 1
Reviewer 1 Report
Comments and Suggestions for Authors
The authors present a descriptive overview of the field of breast reconstruction, formally known as “narrative review”.
While the topic is not original, the readers' great interest make it suitable for consideration
The paper is a reflection of the current situation with the pros and cons of patients who have undergone mastectomy, including the surgical options available, the side effects and limitations of each technique, and the expected outcomes.
This specific type of publication does not require robust statistics or have to prove a predetermined hypothesis, so the value of the work should be measured only on the basis of the balance and organization of the review.
I believe that the authors have provided several interesting insights on the topic. The review is well structured and balanced, and there are no personal opinions reported. Thus, the overall scientific soundness of the work meets the criteria for publication.
The references are adequate in number and pertain to the subject.
The figures on the paper are an effective illustration of the concept of various breast reconstruction methods discussed in the text. They are relevant and of high quality.
I have no further issues to pose, nor do I have to suggest any revisions to the text or language.
If the article's type and subject are within the journal's objective and scope, I think it may be considered for publication.
Author Response
We the authors thank you for your generous appreciation of our manuscript. Although we do not consider it an extensive review of literature but a “narrative" one this manuscript provided us with the opportunity of presenting our initial expertise in this area.
Reviewer 2 Report
Comments and Suggestions for Authors
The authors can elaborate on the following
1. It is not only patients factors and the availabilty of resources when considering breast reconstruction, but oncological factors precluding immediate reconstruction should be discussed
2.When addressing the problems and issues is the impact of complications following a reconstruction on adjuvant treatment and/or survival
3. For implant based reconstruction the long term problems such as capsular contracture, risk of other cancers, needs for revision surgery also needs to be highlighted
4. Factors which have contributed to the rise of contralateral mastectomy such as changes in the guidelines for genetic tests and Angelina Jolie effect in the western population
Author Response
First, we the authors would like to express how deeply honored we are that our work was submitted to such a rigorous and extremely patient reviewer. We are delighted that you have dedicated your time to an in-depth analysis of our manuscript and we thank you for your suggestions, which will allow us to improve our work and produce relevant material. After reviewing our material in accordance with your suggestions, we also found a few additional small mistakes that we were able to rectify in time, and we are grateful for your contribution.
Second, the reviewer made some very reasonable points. They were omissions in our part and we revised the text per his/her suggestions, rewritten and extended some paragraphs. Point by point we addressed the reviewer’s suggestions as follows:
- We discussed in rows 200-211 of the revised manuscript other factors besides patient choice and availability of resources including oncologic factors and flap related conditions.
- And 3. We have addressed these points together by the addition of rows 236-252 of the revised manuscript. Implant based breast reconstruction complications were discussed further highlighting capsular contraction, implant related cancers, aesthetic complications, wound healing delays, hematoma and seroma formation, necrosis. The possibility of additional surgery was pointed out and the oncologic outcomes were mentioned (delaying of adjuvant therapy, overall diminished survivals)
- See above
- We added rows 52 to 59 describing the factors contributing to contralateral mastectomy.